# Novel miRNA Targets and Therapies in the Triple-Negative Breast Cancer Microenvironment: An Emerging Hope for a Challenging Disease

**DOI:** 10.3390/ijms21238905

**Published:** 2020-11-24

**Authors:** Amal Qattan

**Affiliations:** 1Breast Cancer Research Unit, Department of Molecular Oncology, King Faisal Specialist Hospital and Research Centre, Riyadh 11211, Saudi Arabia; akattan@kfshrc.edu.sa or aqattan5@gwu.edu; 2Department of Biochemistry and Molecular Medicine, School of Medicine and Health Sciences (SMHS), George Washington University, Washington, DC 20073, USA

**Keywords:** triple-negative breast cancer (TNBC), microRNAs (miRNAs), tumor microenvironment (TME), cancer stem cells (CSC), exosomes, extracellular vesicles (EVs), cancer therapy

## Abstract

Treatment of triple-negative breast cancer (TNBC) remains challenging because of the heterogeneity of the disease and lack of single targetable driving mutations. TNBC does not rely on estrogen, progesterone or epidermal growth factor receptors and is associated with aggressive disease progression and poor prognosis. TNBC is also characterized by resistance to chemotherapeutics, and response to immunotherapies is limited despite promising results in a subset of TNBC patients. MicroRNAs (miRNAs) have emerged as significant drivers of tumorigenesis and tumor progression in triple-negative breast cancer (TNBC) and present unique opportunities to target various components of the TNBC microenvironment for improved efficacy against this difficult to treat cancer. Effects of miRNAs on multiple targets may improve response rates in the context of this genetically and biologically heterogeneous disease. In this review, we offer a comprehensive view of miRNA regulation in TNBC, treatment challenges presented by TNBC in the context of the tumor microenvironment and stem cell subpopulations, and current and emerging miRNA-based therapeutic strategies targeting various components of the TNBC microenvironment. In addition, we offer insight into novel targets that have potential for treating TNBC through multiple mechanisms in the tumor microenvironment simultaneously and those that may be synergistic with standard chemotherapies.

## 1. Introduction

Breast cancer is the most common malignancy in women and the leading cause of cancer death in women worldwide [1]. There has been a general increase in breast cancer cases worldwide in the past decade [1,2]. Increasing incidence is thought to be related to population growth, an aging population and increased systematic diagnosis, and there were over 2 million new cases and over 600,000 deaths worldwide from breast cancer in 2018 [3]. Three hormonal and growth factor receptors are predominantly used for classification of breast cancers: estrogen receptor (ER), progesterone receptor (PR) and the epidermal growth factor receptor HER2 [4]. Expression of these three molecular markers guides therapy and is correlated with prognosis. Of receptor-positive breast cancers, HER2-positive cancers are the most aggressive but respond well to HER2-targeted therapy [5]. ER or PR positive phenotypes are associated with less aggressive cancers and better survival outcomes [6]. There are numerous estrogen-regulated microRNAs (miRNAs) with functions in proliferation, motility, migration, apoptosis and angiogenesis [7].

Triple-negative breast cancer (TNBC) accounts for 10–15% of breast cancer cases [8] with hormone receptor-negative patients having a 2.6-fold higher risk of mortality [6]. TNBC tumors are highly heterogeneous both between patients and within tumors, making effective treatment difficult [9]. Standard therapy for TNBC is limited mostly to chemotherapy, although this subtype displays particularly chemoresistant phenotypes [10]. Since TNBC is a heterogeneous disease, multiple classification systems have been proposed [10,11,12]. Stromal factors and fibroblasts, immune suppression versus activation [10], tumor microenvironment (TME) classifications [13] and stem components [14] each contribute to tumor aggressiveness, prognosis, chemoresistance and targetability. Each of these components of the TME has unique molecular and genetic regulatory profiles that present different potential therapeutic targeting strategies.

Post-translational regulation of messenger RNA translation by miRNAs is a critical system in tumors and the TME that is known to be dysregulated in TNBC [15]. There is a general down-regulation of miRNA biogenesis in tumor cells, although some miRNAs are up-regulated and play oncogenic roles [16,17,18,19]. The miRNA biogenesis process is initiated by transcription of primary miRNA (pri-miRNA) by RNA polymerase II. De-capped and de-adenylated pri-miRNA is derived from precursor miRNA by DROSHA(RNase)/DGCR8 complex processing, resulting precursor miRNA (pre-miRNA), which is exported out of the nucleus. There, binding and cleavage by the DICER RNase results in mature double-stranded miRNA of approximately 20 nucleotide pairs. The complementary “guide” strand of the mature miRNA serves as a recognition site for the RISC RNase complex to bind and degrade target mRNAs [16].

Dysregulation of miRNAs is common in cancer and occurs through various mechanisms, including amplification/deletion, epigenetic and transcriptional dysregulation and defective miRNA biogenesis [20]. These dysregulated miRNAs can act as tumor suppressors or oncogenes. For example, miR-15 and miR-16a can suppress tumors through apoptosis by inhibiting Bcl-2 expression [21]. As an oncogene, miR-663 was found to target p21WAF/CIP1, which promoted cell cycle progression to promote nasopharyngeal carcinoma tumorigenesis [22]. Many miRNAs have been found to affect drug resistance in cancer though multiple mechanisms, including regulation of ABC transporters, apoptosis and the DNA damage response [23]. Secreted miRNAs found in the circulation are now known to aberrantly accumulate in cancers reflecting both pathologic conditions and selection of miRNAs for packaging into extracellular vesicles [24]. Thus, circulating miRNAs have garnered high interest as diagnostic, prognostic and theranostic markers of cancer.

MiRNA regulators are both markers and determinants of the pathobiology of TNBC [15,25]. In the context of the tumor microenvironment, miRNAs are intimately involved in processes of epithelial to mesenchymal transition (EMT), secretion from fibroblasts, inflammation, survival, target expression and stemness [16]. Targeting either dysregulated or functional miRNAs in various components of the TME, including stem cells, is a novel and attractive approach to advanced therapy for TNBC [25,26]. Since miRNAs broadly alter these functions of the TME that are common pathobiologic factors in an otherwise heterogeneous disease, miRNA-based approaches may offer better response rates in this intractable tumor type. Specific differences exist between miRNA expression profiles in TNBC and ER-positive breast cancer. For example, miR-222 and miR-221, which directly target ERα transcripts, are more highly expressed in ER-negative tumor cells [23].

The goal of this review is to clearly present the context of miRNA function in TNBC, recently discovered roles of miRNAs and emerging potential miRNA therapeutics targeting TNBC microenvironmental components. We begin with a summary of the context of TNBC classifications and relevant components of the TNBC microenvironment that can specifically influence and be acted upon by miRNAs.

## 2. Challenges in TNBC Therapy

### 2.1. Heterogeneity of TNBC

Breast cancer is classified into five subtypes including HER2-enriched, luminal-A, luminal-B, basal-like and normal breast-like [27,28,29]. The most common classical subtype within TNBC is basal-like [11,27,28,30]. Basal-like tumors express cytokeratins (5/6 and 17) but typically do not express ER/PR/HER2 [27]. While 50–75% of TNBC can be described as basal-like using the intrinsic breast cancer classification system [11,27,30], there is considerable molecular and phenotypic variation within TNBC. Response to neoadjuvant therapy can be predicted using a 50-gene expression panel (PAM50) to classify breast cancers into these intrinsic subtypes [31]. More recently, the understanding of the complexity of molecular subtyping within TNBC has been expanded. Pronounced inter-patient heterogeneity within TNBC therefore presents a challenge in finding effective chemotherapy or targeted therapy.

The heterogeneity of TNBC has been demonstrated by genetic and epigenetic analysis by The Cancer Genome Atlas and other groups [32,33,34]. These studies revealed that approximately 36% of basal-like tumors harbored inactivation mutations in BRCA1 or BRCA2. TP53 mutation was quite common in basal-like and HER2-enhanced subtypes, comprised mostly of TNBC; however known driver mutations were variable and not found in 12% of TNBC [32,34]. Alterations in PI3K/AKT pathway signaling also occurred in a small subset of basal-like breast cancers (mutated in only 9% and PTEN loss in 35%) [33]. PTEN has been shown to be targeted by miRNAs, e.g., miR-21 and miR-222, in TNBC, affecting tumorigenesis and drug resistance as discussed below. PI3K mutations have recently been associated with poorer response to anthracycline treatment of TNBC [35]. These studies also implied molecular heterogeneity among TNBCs from the point of tumor initiation and that TNBCs have a low mutational burden [34], which bodes poorly for the potential of immunotheraties in this disease.

TNBC has been classified into six subtypes based on gene expression profiles including basal-like 1 (BL1) and basal-like 2 (BL2), immunomodulatory (IM), luminal androgen receptor positive (LAR), mesenchymal (M) and mesenchymal stem-like (MSL) [32,36]. Each of these subtypes has potential therapeutic vulnerabilities, including sensitivity to the androgen receptor agonist bicalutamide in the LAR group [36]. In 2015, Burstein et al. presented a model of classification based on genomic profiling with distinct prognoses among subtypes [12]. The subtypes resulting from this analysis include luminal androgen receptor-positive (LAR), mesenchymal (MES), basal-like immune suppressed (BLIS) and basal-like immune activated (BLIA). The MES subtype includes “claudin-low” tumors that lack markers of luminal differentiation and have a poor prognosis compared to other intrinsic breast cancer subtypes and stem-like phenotypes [12,29,37]. Therapies targeting stem-markers or functional effectors of stem-like phenotypes may hold promise in treating the MES subtype. Within this classification system, immune-activation had the strongest association with a positive prognosis, and immune suppressed basal-like cases had the worst prognosis [12]. Immune activated phenotypes may be more responsive to immune checkpoint inhibitors (ICIs). These subtypes represent identifiable tumor profiles, each having a unique set of potential therapeutic targets that can be regulated by global systems, including miRNAs. The Burstein classification system, its intersection of basal-like breast cancer and TNBC and therapeutic opportunities are illustrated in Figure 1. The unique pathobiology of these heterogeneous subtypes and treatment implications are becoming better understood through investigation of miRNA regulation in these contexts. The relevance of specific miRNAs may vary according to breast cancer phenotype. For example, selected relevant miRNAs are presented in Figure 1. The potential implications of subtypes classified according to such tumor intrinsic and microenvironmental factors and their respective prognoses and therapeutic targets in the context of miRNAs will be discussed in more detail below.

### 2.2. Conventional and Emerging Therapies for TNBC

Before the emergence of poly ADP-ribose polymerase (PARP) inhibitors and immune checkpoint inhibitors (ICIs), a variety of chemotherapies were recommended for advance breast cancer [38]. Anthracyclines cause DNA damage and inhibit replication by blocking the relaxation of supercoiled DNA by topoisomerase and can be associated with cardiotoxicity [39]. Anthracyclines have become standard for neoadjuvant treatment of breast cancer, although an overall survival benefit of treatment regimens including the anthracycline doxorubicin compared to those not including anthracycline (docetaxel and cyclophosphamide only, TC) [37,40]. Anthracycline use may be particularly ineffective in patients with impaired PI3KCA [35].

### 2.3. Chemoresistance Mechanisms

TNBC is generally more immunogenic than HER2-positive breast cancer with the presence of immune infiltrates and a somewhat higher mutational load [37,41]. The programmed death-ligand 1 (PD-L1) is expressed on immune infiltrates in 40–65% of TNBC which can be predictive of response to immune checkpoint inhibitors [37,42,43,44].

The heterogeneity in molecular profiles among TNBC cases, interactions within the TME, drug efflux, survival signaling, hypoxia-induced angiogenesis and EMT and cancer stem cell phenotypes each play roles in resistance to both chemotherapy and targeted therapies in TNBC [45]. Many of these aspects of triple-negative breast tumors can be regulated by miRNAs, both intrinsically and through therapeutic miRNA modalities, as will be discussed in more detail in Section 4 below.

Mechanisms behind chemoresistance associated with EMT and stemness are controversial, but have been hypothesized to include quiescence, resulting in evasion of replication-dependent genotoxicity, and immune evasion [26,46]. Wnt signaling, which promotes stemness and tumor initiation, can be dysregulated in TNBC [45,47]. Aberrant Wnt signaling in TNBC is associated with increased metastatic potential and can be regulated by miRNAs in breast cancer [48,49,50,51,52]. Hypoxia can cause enrichment of breast cancer stem cells (BCSCs) mediated by hypoxia inducible factor 1 alpha (HIF1α) [26,52]. This limits the efficacy of chemotherapeutic agents, antitumor immunity, antiangiogenic drugs [52]. In fact, HIF1α was found to be required for multi-drug resistance to paclitaxel and gemcitabine in human TNBC stem cell populations [53].

Membrane transporters play a role in multi-drug resistance in many cancer types by mediating drug and reactive oxygen species (ROS) efflux [26,45,54]. For example, the P-glycoprotein transporter (P-gp) can mediate resistance to doxorubicin in TNBC [54]. This effect was found to be secondary to decreased intracellular accumulation of reactive oxygen species and downstream inhibition of an Akt/NF-κB signaling axis. The other major drug efflux pump involved in multiple drug resistance in breast cancer is the human breast cancer resistance protein (BCRP/ABCG2) [55].

There are also multiple immune and stromal determinants of chemoresistance within the microenvironment of TNBC. Cancer-associated fibroblasts (CAFs) secrete extracellular matrix proteins and factors that can block drug delivery to tumor cells [56] and increase resistance in cancer stem cells through STAT/Notch signaling stimulated by fibroblast exosomal RNA [57]. Increased expression of the 5′-nucleotidase CD73, which hydrolyses ATP to ADP, has been found to confer resistance to the mainstay chemotherapeutics in TNBC, anthracycline, through suppression of adaptive immunity [58]. Mesenchymal stem cells (MSC) in the TME can interact with tumor cells to increase resistance to immune checkpoint inhibitors through Src/PI3K/Akt signaling and enhanced survival [59]. Endothelial cells also induce NF- κB signaling in cancer stem cells via secreted TNFα, also creating a cytokine loop with immune cells and resulting in resistance to doxorubicin and cyclophosphamide [26].

Each of these aspects of TNBC affect and are affected by miRNA functions and expression. Each presents unique opportunities to exploit miRNAs as diagnostic, prognostic or theranostic markers, targets for therapy, modalities of therapy and tools for understanding the pathobiology of TNBC. A detailed discussion of specific miRNAs that have been recently revealed to play roles in these aspects of TNBC and their implications on the understanding of TNBC pathobiology and therapy follows.

## 3. The TNBC Microenvironment

The interplay between tumor cells, the extracellular matrix (ECM), stromal normal and activated fibroblasts and immune infiltrates is well known to affect tumor growth, progression and response to therapy. The components of the TNBC microenvironment and the processes within each that affect TNBC pathobiology are discussed below and summarized in Figure 2. Here we present an overview of the TNBC microenvironment, which sets the context for the subsequent discussion of miRNA-regulated mechanisms within this microenvironment that are relevant to tumorigenesis, tumor progression and therapy.

### 3.1. Immune Cells

Among breast cancers, TNBC has an increased prevalence of immune infiltration at the invasive margin that is primarily composed of tumor-infiltrating lymphocytes (TILs), in particular T cells [60,61,62,63]. Higher infiltration of TILs is associated with a higher pathologic complete response in TNBC [60,64]. CD8 cytotoxic T cells are primarily involved in tumor killing and improved prognosis [65]. Conversely, CD4+CD25+FoxP3+ regulatory T cells are immunosuppressive and correlate negatively with prognosis in TNBC [66,67]. A more robust correlation of TILs with prognosis may be related to relatively higher levels of chromosomal instability and mutational burden in TNBC compared to other breast cancer subtypes [37,41,68].

The presence of myeloid-derived suppressor cells and tumor-associated macrophages (TAMs) are generally considered to be associated with poor prognosis [69,70]. Macrophages are generally divided into M1 and M2 subtypes [60,71]. While M1 macrophages are pro-inflammatory and thereby anti-tumorigenic, M2 (alternatively activated) macrophages secrete growth factors and cytokines that promote tumor growth [72]. TNBC tumor cells secrete factors, including macrophage colony stimulating factor (M-CSF) and IL-6, that drive macrophages toward M2 polarization [60,73,74]. Accordingly, CD163+ and CD68+ M2 macrophages are more abundant in TNBC/basal-like breast cancer than luminal types [74]. While CD68+ macrophages secrete IL-6 and CCL5, which correlate with poor prognosis, the prognostic significance of TAMs in the TNBC microenvironment is not well delineated [60,75].

### 3.2. Stromal Components

Normal fibroblasts are activated into myofibroblast-like cancer-associated fibroblasts, which support tumor growth and invasion [76]. CAFs promote tumor progression, angiogenesis, inflammatory infiltration, invasion and metastasis [77,78]. Activated CAFs promote TNBC cell metastasis through expression of matrix metalloproteinase 9 (MMP-9) expression [79]. Basal-like breast cancer tumor cells interact with CAFs to induce expression of chemokines and interleukins [80]. This interaction was associated with increased tumor cell migration. Vascular endothelial growth factor (VEGF) expression is significantly altered in favor of angiogenesis in TNBC [81], and VEGF expression is significantly higher in TNBC that in other breast cancer subtypes [82]. VEGF is known to increase tumor cell adhesion and invasion of TNBC tumor cells across the surrounding endothelium [83]. The high expression of VEGF is associated with poor prognosis in TNBC patients and is correlated with increased lymph node metastasis and poorer overall survival [84].

The extracellular matrix (ECM) is composed of structural proteins, glycoproteins and proteoglycans with a variety of regulatory and structural functions. The main structural proteins, type-I and type-IV collagen, are responsible for physical barrier and basement membrane functions, respectively, with the latter influencing tissue polarity [60]. In TNBC cells, type-I collagen has been found to increase tumor aggressiveness and induce MMP-9 expression [85]. Collagen cross-linking and fibronectin assembly, and therefore, stromal stiffness, are induced by lysyl oxidase (LOX) [86]. Stromal stiffening is associated with increased drug exclusion, tumor aggressiveness and TAM infiltration [86,87]. Collagen I production, stiffening and these downstream tumor and TME phenotypes were induced by TGFβ signaling in breast tumor cells [87]. Loss of mechano-chemical signaling through the interaction of integrins with type-IV collagen in the basement membrane results in EMT, loss of polarity and migration [88]. However, native type-IV collagen has been shown to increase migration, MMP secretion and invasion in TNBC cells via CD9 and discoidin domain receptor 1 (DDR1) [89,90].

### 3.3. Cancer Stem Cells

In addition to being resistant to drugs, cancer stem cells are self-renewable and a source of protracted tumor cell proliferation and metastasis [46,91]. Cancer stem cells are an important therapeutic target given their role in metastasis, residual disease and relapse. In 2003, the first description of breast cancer stem cells (BCSCs) [92] and the importance of Wnt signaling in the establishment of BCSCs [93] were reported. Self-renewal is related to embryonic development pathways, including Wnt/β-catenin, NOTCH and Hedgehog pathways [46]. Wnt has been found to be important for metastasis of breast cancer as well [94]. The accumulation of β-catenin is pronounced in TNBC cells compared to non-TNBC breast cancer cells, which marks Wnt signaling and stemness [14,95]. BCSCs are marked by expression of ALDH, CD133 and CD44. CD44 can be regulated by hypoxia inducible factor-2α (HIF2 α) in TNBC and plays a key role in the maintenance of stem-like phenotypes and migration through activation of the PI3K/Akt/mTOR pathway [96]. Aldehyde dehydrogenase(ALDH) expression is associated with increased metastasis [97], drug resistance [98] and radiation resistance [99] in TNBC cells. Expression of CD133 in TNBC is correlated with EMT markers, poor prognosis and resistance to neoadjuvant chemotherapy and plays a functional role in promoting invasion in TNBC [100]. While stem cells represent important pathobiologic components of TNBC tumors and are therapeutically challenging, they also have unique epigenetic and transcriptional profiles that present potential targets for therapy, including that using broad post-transcriptional regulation using inhibitory RNAs.

## 4. miRNAs in Triple-Negative Breast Cancer

### 4.1. miRNAs in TNBC Tumorigenesis, Promotion, Progression and Stemness

Regulation by miRNAs has been strongly implicated in the progression of TNBC. Lymph node metastasis and a basal subtype was found to be associated with breast tumor expression of miR-155, miR-320a and miR-205 [13]. These associations extended to circulating miRNAs as well. Migration and invasion of TNBC cells has been found to be affected by miR-206 through negative regulation of the tetraspanin family member transmembrane 4 L-six family member 1 (TM4SF1), which is involved in cell growth and motility [101]. Invasion and metastasis can also be inhibited by miR-340 in TNBC cells through inhibition of Rho Kinase 1 (ROCK1) [49]. The well-known breast cancer oncogenic signaling protein STAT3 is down-regulated by the tumor-suppressive miR-124 in TNBC cells resulting in the inhibition of cell proliferation and invasion [102]. It is reasonable to hypothesize that tumor evolution subsequent to tumor initiation is driven by selection of miRNA post-translational regulatory programs that favor proliferation, migration, invasion and metastasis. The prognostic implications of such tumor progression supporting programming is reflected in the discovery of a panel of miRNAs with prognostic predictive value in TNBC [103]. Specifically, the expression of eight miRNAs (miR-139-5p, miR-10b-5p, miR-486-5p, miR-455-3p, miR-107, miR-146b-5p, miR-324-5p and miR-20a-5p) was found to predict post-surgical relapse in TNBC patients. Of these, miR-139-5p, -10b-5p and -486-5p were down-regulated and the remaining miRNAs were up-regulated in TNBC as analyzed using The Gene Expression Omnibus (GEOD) and The Cancer Genome Atlas (TCGA) data. Of all associated miRNAs, miR-139-5p showed the strongest correlation with disease free survival and TNM stage, although all associations with overall survival were weaker. Tumor suppressive function of miR-139 may involve regulation of targets including VEGFR, IGF-1R, HOXA10, Wnt, Ras, PI3K, NFκB and ROCK2 [103,104]. LOX, which is involved in collagen crosslinking, is targeted by miR-142.3p along with HIF1α and integrin subunit alpha 5 (ITGA5), thereby overcoming chemoresistance in TNBC [86]. PTEN has been shown to be targeted by miR-21 in TNBC cells, resulting in the promotion of proliferation and invasion [105]. Understanding miRNA regulatory pathways and their direct roles in tumor progression and metastasis provide a powerful knowledge base on which to build therapeutic strategies for TNBC.

Given the ability of cancer stem cells to self-renew and metastasize, miRNA regulation of the processes of EMT, self-renewal, invasion and metastasis in BCSCs is of particular interest. The Wnt pathway and its roles in the malignancy-related features of breast cancer stem cells are inhibited by miR-340 and miR148a, particularly affecting migration, invasion and metastases in TNBC cells [25,48,50]. Wnt/β-catenin target genes and stem markers can be down-regulated by miR-137, through repression of expression of the Follistatin like 1 (FSTL1) secreted glycoprotein [51]. The miR-200 family(-200a, b and c) play positive roles in stem-like features of BCSC, including those related to metastasis, EMT, survival and growth [25,106]. Targets of miR-200 are outlined in Table 1. EMT and stemness are inhibited by monocarboxylate transporter-1 (MCT-1)-induced miR34a in TNBC through IL-6R depletion [73]. The miR-199a/217 cluster was found to be under-expressed in TNBC and to negatively regulate EMT and cell cycle progression [107]. Finally, miR-4417 was found to be tumor suppressive and inhibit the stem-like property of mammosphere formation in TNBC cells [108].

### 4.2. miRNAs in TNBC Drug Resistance

There are a number of miRNAs that have been found to regulated resistance to chemotherapeutic agents among breast cancer types. These include miRNA-222, which can confer doxorubicin resistance through exosomal transfer [109], which will be described in more detail in Section 4.4. Several miRNAs have been found to be involved in the response of TNBC to chemotherapeutic agents. For example, miR-5195-3p was shown to cause sensitization to paclitaxel in TNBC through depletion of EIF4A2 transcripts [110]. Doxorubicin resistance can be mediated by the miRNA-449 family through cell cycle regulation [111]. In addition to its effects on Wnt signaling and stemness, the miR-137 target mediates multi-drug resistance in TNBC cells [51]. An association has been found between docetaxel resistance in human breast cancer cells with miR-34a, including that in TNBC [112], while specific mechanisms remain unresolved; however, in the context of EMT and stemness, miR-34a acts as a tumor suppressor and may confer drug resistance, or at least resistance to docetaxel. It was proposed that this effect may have been mediated by targeting of Bcl-2 and Cyclin D1 by miR-34a [112].

### 4.3. miRNA Regulation of the TNBC Microenvironment

Since the function of many components of the TME affect tumor growth and progression in TNBC, it is important to consider miRNA regulation in this context. Cellular processes within the TNBC microenvironment are summarized in Figure 3 and discussed below.

#### 4.3.1. miRNAs in Fibroblasts and CAFs

As integral parts of the TME that can exhibit tumor supportive interactions as discussed in Section 3.2, post-transcriptional regulation programming in fibroblasts and cancer-associated fibroblasts in the TNBC microenvironment is relevant to the pathobiology and treatment of the disease. Loss of the negative regulator of PI3K/Akt signaling PTEN in stromal fibroblasts leads to miR-320-mediated depletion of E26 oncogene homolog 2 (ETS2) [113]. Targeting of ETS2 by miR-320 and a resulting activation of an oncogenic secretome that promotes angiogenesis and invasion in tumor cells was found to be a critical mechanism for these phenotypes in PTEN-deleted fibroblasts [113]. This secretome included matrix metalloproteinases. Yes-associated Protein 1, which regulates proliferation and differentiation, is targeted by miR-205 in breast fibroblasts converting them to activated CAFs [114]. This was also found to promote angiogenesis through a VEGF-independent pathway. CAFs demonstrate increased miR-222 expression relative to normal fibroblasts, which was found to increase migration, invasion and senescence and lead to CAF activation through its depletion of lamin B receptor (LBR) [115]. Regulation of LBR-induced senescence and induction of EMT were mechanisms that were associated with the miR-222–LBR axis. This led to enhanced TNBC cell migration and invasion.

#### 4.3.2. miRNAs in Immune Infiltrates

In addition to its effects on stemness of BCSCs, miR34a can inhibit M2 macrophage polarization and favors anti-tumorigenic M1 polarization, again through depletion of IL-6R [73]. With the knowledge that M2 polarization is actively promoted by TNBC tumor cells and has negative prognostic significance, the miR-34a-macrophage pathway becomes important to the pathobiology of TNBC with potential therapeutic implications, which are discussed in further detail in Section 5. PD-L1 expression, which mediates immune evasion, is a direct target of miR-195/miR497 in TNBC [116]. Circulating miR-195-5p, which is up-regulated in TNBC [117], has also been identified as a candidate regulator of PD1 in TNBC [118]. Another circulating miRNA in TNBC, miR-155 [119], is known to regulate the immune checkpoint receptor CTLA4 [118].

An interesting immunologic phenomenon that is linked to cancer progression with emerging interest in the breast cancer field is neutrophil extracellular trapping (NET or NETosis). NETosis occurs when activated neutrophils decondense their chromatin and expel a network of DNA and cellular constituents from the cell [120]. Cancer cells can promote NETosis by expression of granulocyte-colony-stimulating factor (G-CSF) and interleukin-8 (IL-8) [120]. These NETs can promote venous thromboembolism and catch tumor cells promoting metastasis [121,122]. Regulation of NETosis by miRNAs has been implicated by pathway analysis of differentially expressed miRNAs, including miR-499, miR-20b, miR23b and miR1298-5p, in breast cancer cells treated with anti-tumorigenic anacardic acid [123]. With little definitive data in the literature, further research is justified to evaluate the role of miRNAs in regulation of NETosis in breast cancer and its implications on disease progression.

### 4.4. Extracellular Vesicles (EVs) and Exosomal miRNAs in the TNBC Tumor Microenvironment

Exosomes, extracellular vesicles (EVs) carrying protein, DNA and RNA, can transfer miRNAs with effects on surrounding TME components. In fact, breast cancer exosomes have been found to contain miRNAs complexed with RISC which can be processes outside the context of cells [124,125]. These exosomes contain pre-miRNAs, DICER, AGO2 and TRBP for mediation of miRNA processing [125]. Exosomes released from breast cancer cells may influence drug resistance in other tumor cells, oncogenic transformation of normal breast epithelial cells and have immunomodulatory effects in the TME [109]. Drug resistance, including that to adriamycin, has been found to be transferred from resistant breast cancer cells by miR-222 in exosomes [109,126,127]. Drug resistance conferred by miR-222 may, in part, be due to the targeting of PTEN [128]. Likewise, exosomal transfer of miR-155 from resistant TNBC cells can confer multi-drug resistance [109,129]. Transferred resistance in this context was associated with promotion of EMT [129]. Docetaxel treatment of TNBC cells can induce miR-9-5p, miR-203-3p and mir-195-5p content in extracellular vesicles, which target one cut homeobox 2 (ONECUT2) transcription factor to promote stemness and resistance [130].

Pro-metastatic miR-9 is up-regulated in TNBC tumor cells and transferred in exosomes to fibroblasts [131]. This leads to the activation of normal fibroblasts into CAFs through regulation of cell motility and extracellular matrix remodeling genes, thereby promoting tumor growth and invasion. Similarly, miR-125b, which is highly secreted by TNBC tumor cells in extracellular vesicles (EVs), is transferred to fibroblasts causing their activation and CAF phenotypes through targeting of TP53-inducible nuclear protein 1 (TP53inp1) [132]. TNBC cells likewise express high levels of miR-1246 and secrete them in exosomes, which is transferred among cellular components of the TME resulting in depletion of the target gene CCNG2 and tumor cell viability, migration and chemoresistance [133]. Mesenchymal stem cells release miR-100 in exosomes, which modulate mTOR/HIFα to decrease VEGF expression and secretion by tumor cells in the breast cancer TME resulting in decreased angiogenesis [134,135].

## 5. miRNA Therapeutics Targeting TNBC Microenvironmental Components

Ongoing and recent discovery of miRNA regulatory networks and their roles in the pathobiology and drug resistance of TNBC continue to present new RNA inhibition modalities for the treatment and chemosensitization of this relatively intractable disease. The use of miRNAs as therapy for breast cancer is an emerging field, and the vast majority of the current literature in this field is in the preclinical stage. Below, select targets, approaches and delivery systems for exploiting miRNA regulation in TNBC are discussed.

### 5.1. Targeting the Tumors and the Tumor Stroma

The role of hypoxia-induced lysyl oxidase (LOX) on the tumor microenvironment and chemoresistance has been targeted using stable exogenous re-expression of miR-142-5p [86]. High LOX expression was shown to be associated with poorer survival in TNBC patients, particularly in those who were treated with chemotherapy. Hypoxia inhibited miR-142-5p expression and increased expression of LOX in TNBC cells. Stable re-expression of miR-142-5p robustly and significantly increased sensitivity of TNBC cells embedded in collagen to doxorubicin. The delivery of this miRNA to drug resistant TNBC in patients in combination with chemotherapy may be a promising therapeutic approach.

Transfection of miR-206 in TNBC cells was shown to downregulate TM4SF1, which was associated with migration and invasion as reported by Fan et al. in 2019 [101]. This study showed that knockout of TM4SF1 or transfection of miR-206 reduced tumor growth in TNBC xenografts. Maskey et al. (2017) showed that overexpression of miR-340 in TNBC cells achieved down-regulation of ROCK1, which mimicked the effects of direct ROCK1 silencing on inhibition of migration, invasion and proliferation [49]. Each of these miRNAs represents a potential therapeutic for TNBC.

### 5.2. Targeting Immune Components of the TME

Targeting the immune microenvironment may be an effective means to tip the balance of TNBC more heavily toward an immune activated or “hot” status. Increasing the infiltration and activation of cytotoxic T-cells and avoidance of tumor-promoting myeloid infiltrates, including M2 macrophages is a well-studied and validated approach to potentiating natural immune killing of tumors as well as immunotherapies, such as immune checkpoint inhibitors. There is evidence that miRNAs may provide an effective modality for such therapy. Decreased miR-34a expression downstream of MCT-1 in TNBC cells was found to suppress IL-6R expression and suppress M2 differentiation of co-cultured macrophages [73]. Exogenous expression of miR-34a was shown to reverse this effect.

### 5.3. Targeting Cancer Stem Cells

In addition to affecting macrophage differentiation, miR-34a may directly affect stemness and metastasis of TNBC. Delivery of telomerase (TERT) promoter-driven miR-34a expression using nanoparticles was shown to abrogate the long-term maintenance of TNBC stem cells by depleting C22ORF28 (chromosome 22 open reading frame 28) [136]. This study determined that C22ORF28 expression is frequently higher in human breast tumors and that high C22ORF28 or low mir-34a expression was associated with poor survival in human breast cancer. Subsequently, Weng et al. (2019) demonstrated that miR-34a repression by monocarboxylate transporter-1 (MCT-1) promotes stemness in TNBC cells and that this phenotype was associated with the targeting of IL-6R [73]. Transfection with miR-34a reversed this phenotype.

Anti-miRNA technology has also been investigated in the targeting of TNBC stem cells. Yin et al. (2019) utilized RNA nanoparticles to target CD133 on the surface of TNBC stem cells and deliver an anti-miR-21 therapeutic [137]. CD133 binding and specificity to BCSCs was conferred by an RNA aptamer and a locked nucleic acid antisense inhibitor of miR-21. Delivery of anti-miR-21 by this methodology inhibited expression of miR-21 and up-regulated expression of target tumor suppressors PTEN and PDCD4. Intravenous injection of nanoparticles inhibited tumor growth in TNBC xenograft models.

### 5.4. Targeting miRNAs

In addition to direct use of miRNAs to target TNBC, there is rationale and evidence to support the use of epigenetics to target miRNAs in breast cancer. Epigenetic regulation is implicated in dysregulation of miRNAs in cancer [138]. Histone deacetylase inhibitors have been shown to effectively inhibit breast cancer cells through miRNA regulation. For example, Bian et al. (2018) demonstrated HDAC inhibitor suppression of TNBC cell proliferation and invasion via regulation of miR-200c expression, which in turn targets the Crk-like (CRKL) adapter protein [139]. There is sound rationale for targeting one or more miRNAs to broadly affect expression of cancer-related genes and thereby affect multiple tumor suppressive or promoting pathways in TNBC and other cancers. This approach may be less prone to acquired resistance and better efficacy because of the use of one target to affect multiple pathways.

### 5.5. Exogenous miRNAs

An interesting topic that has recently been investigated is the use of non-human miRNAs on human cancer-related targets. The use of plant miRNAs that can be found in human sera, such as miR-159, which was found to be decreased in breast cancer and inversely correlated with progression, has even been investigated [140]. The T-cell factor 7 (TCF7) Wnt signaling transcription factor was targeted in mouse TNBC xenografts by systemic delivery of a miR-159 mimic [140]. In 2019, that research was extended to show that exosomal miR-159 in combination with doxorubicin silenced TCF7 and enhanced the efficacy of doxorubicin in a TNBC xenograft model [141].

### 5.6. Clinical Studies in miRNA Therapies and Biomarkers

While miRNA therapies for breast cancer remain in preclinical stages, proof of principal has been achieved for inhibitory nucleic acid-based drugs in several other diseases. In cancer, multiple clinical trials have been and currently are being conducted for miRNA therapies and biomarkers [142]. Golan et al. demonstrated, in a Phase 1/2 trial, that siRNA targeting mutant KRAS(G12D) was well tolerated with potential efficacy in locally advanced pancreatic cancer [143]. Controlled efficacy studies have not yet been published on this modality. In breast cancer, some clinical studies have demonstrated the utility of miRNAs as theranostic biomarkers. Decreased miR-4465 was shown to be associated with reduced tumor proliferation in patients being treated with bevacizumab in combination with chemotherapy [144]. VEGF and HER2/HER3 signaling, cell cycle regulation and the DNA damage response were correlated with miR-4465 in the tumors of these patients. As such studies emerge, rationale is strengthened for the clinical utility of miRNA therapy in combination with targeted or chemotherapeutic drugs and the use of miRNAs as biomarkers in breast cancer.

## 6. Conclusions

Either the targeting of miRNAs or the use of miRNAs as therapeutics are promising new potential modalities for the treatment of TNBC. Mounting evidence suggests that multiple aspects of the TME must be considered to improve outcomes in TNBC treatment. It is conceivable that multiple processes occurring in the TNBC microenvironment could be targeted at once using single miRNAs. For example, we have summarized the effects of miR-34a on tumor stemness, drug resistance and polarization of the immune microenvironment toward an immunogenic and anti-tumor status. Furthermore, regulation of these multiple aspects of the TME may be through multiple target transcripts. It is reasonable to hypothesize that manipulation of miRNA regulation of drug-resistance in combination with chemotherapy may improve therapeutic indices and demonstrate synergistic effects. This promises to improve survival outcomes and minimize adverse effects of chemotherapy.

## Figures and Tables

**Figure 1 ijms-21-08905-f001:**
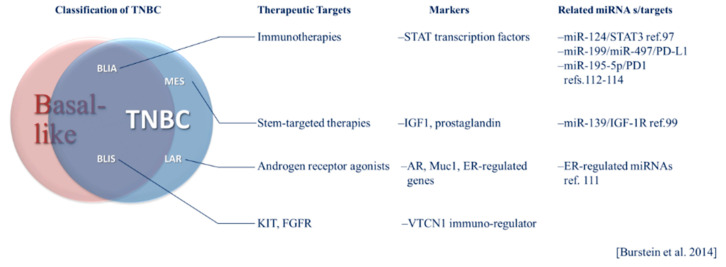
The Burstein system of triple-negative breast cancer (TNBC) classification and its relationship to basal-like breast cancer. General targeted therapeutic considerations, molecular markers and microRNA (miRNA) specific to TNBC subtypes are listed.

**Figure 2 ijms-21-08905-f002:**
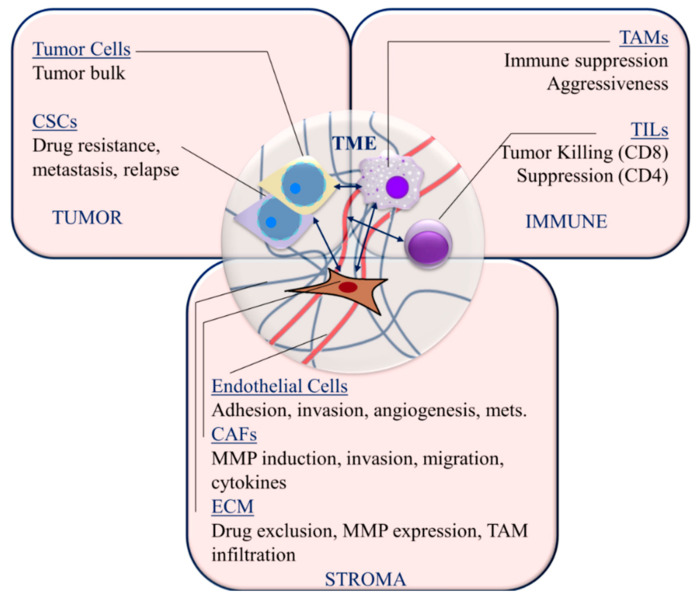
Targetable TNBC microenvironment components and interactions. Cancer stem cells (CSCs); tumor-infiltrating lymphocytes (TIL); tumor-associated macrophages (TAM); cancer-associated fibroblasts (CAFs); matrix metalloproteinases (MMPs); extracellular matrix (ECM); metastasis (mets); mesenchymal stem cells (MSCs). Arrows indicate interactions between cellular components of the tumor microenvironment.

**Figure 3 ijms-21-08905-f003:**
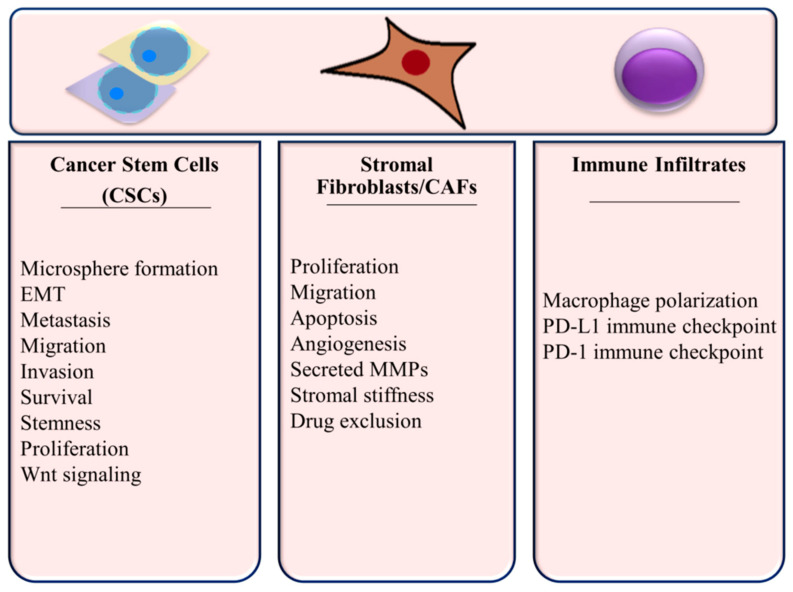
Cellular Processes Regulated by miRNAs in the TNBC Microenvironment.

**Table 1 ijms-21-08905-t001:** Select miRNAs in the TNBC Microenvironment.

TME Compartment	microRNA	Targets	Processes	Publication
Cancer Stem Cells				
	miR-21	PTEN	Promotes proliferation, invasion	[105]
	miR-200	KLF4, EZH2, BMI1, SUZ12, BMI1	promotes microsphere formation, EMT, metastasis, invasion, survival, growth	[106]
	miR-206	TM4SF1	migration, invasion	[101]
	miR-34a	IL-6R	inhibits EMT, stemness	[73]
	miR-199a/214	EMT and ECM targets	inhibits EMT, proliferation, invasion	[107]
	miR-148a	Wnt-1	inhibits Wnt signaling	[50]
	miR-340	Rock1, cMyc, CTNNB1	inhibits Wnt signaling	[48]
	miR-137	FSTL1	Wnt signaling	[51]
	miR-4417	multiple	Inhibits mammosphere formation	[108]
	miR-142-3p	HIF1α, LOX	stromal stiffness, drug exclusion	[86]
Fibroblasts/CAFs				
	ER-regulated miRNAs	multiple	proliferation, migration, apoptosis, angiogenesis	[7]
	miR-320	ETS2	inhibits MMP secretion, angiogenesis	[113]
	miR-205	VEGF-A, ZEB1, YAP1	angiogenesis	[114]
	miR-9	E-cadherin	activation	[131]
	miR-125b	TP53inp1	activation	[132]
	miR-222	LBR	activation, migration	[115]
Immune Infiltrates				
	miR-34a	IL-6R	macrophage polarization	[73]
	mir-195/miR-497	PD-L1	immune checkpoint	[116]
	miR-195-5p	PD1	immune checkpoint	[118]
	miR-155	CTLA4	immune checkpoint	[118,119]

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
