# Peer review of "Novel miRNA Targets and Therapies in the Triple-Negative Breast Cancer Microenvironment: An Emerging Hope for a Challenging Disease"

_ijms, 2020, doi:10.3390/ijms21238905_

Round 1

Reviewer 1 Report

Amal Qattan proposes a review entitled “novel miRNA targets & therapies in the triple-negative breast cancer microenvironment: an emerging hope for a challenging disease. Despite an interesting topic, when we read the paper, the subject appeared too large and the goal of the author unclear (e.g. line 82-85 are unclear). I regret that in the present form, this review is hardly difficult to understand and the molecular pathways played by the miRNA in the breast cancer, that I guess, are the main interest of the review are far from being exploited enough.

Major points:

The review is artificially divided in sections concerning general hallmarks of breast cancer and sections with miRNA related to these cancer phenotypes. There are many references to another section in the text, therefore the logic and the roles of these miRNA are very difficult to understand. For each section the author gave only a quick overview of the subject and the reader is not able to understand whether such information is needed or other is lacking for the understanding of specific miRNA roles in BC etiology. For example, it is unclear for me, whether line 212-220 concerning M1 and M2 macrophages are useful for the purpose of miRNA in BC.

It would be much better to fuse the different sections and keep only that is essential in the descriptive breast cancer related phenotypes for the understanding of miRNA roles in these phenotypes (BC related phenotypes have been often previously described in details in many different reviews).

In my opinion, the goal of this review should be to explain the molecular mechanisms played by the miRNA and how it results in such phenotype. These mechanisms are not fully described. For example, line 356 “miR-320-inhibited pathway was shown to drive tumor-associated fibroblast phenotype…” How? which genes are regulated directly ? this question should be asked for each miRNA cited. May be, the author should focus the review on one specific cancer related phenotype and explore it in details.

miRNA are not present in the different pictures related to BC phenotypes and should be added to better understand their molecular roles.

Other points:

Line 38: general increase of BC incidence is also due to an increase of systematic diagnosis

Some references are under brackets and other not

Line 92: “50-70% of TNBC …basal like” and line 120 “70 percent of TNBC is basal-like”

Line 149 “regimens” what does it mean?

English typo are present; the paper should be edited for English

Reviewer 2 Report

Thank you much for inviting me to review this very comprehensive and very good review on

" Novel miRNA Targets and Therapies in the Triple-Negative Breast Cancer Microenvironment: An Emerging Hope for a Challenging Disease"

Topic is highly actual, interesting and with large implications in clinical practice. The role of miRNA in human cancer is emerging as a very important topic in the research in the last years.

Manuscript is very clearly drafted even for somebody outside molecular research. Mechanisms of action of miRNA in TNBC are well, clearly and comprehensively described.

However, I would like to suggest some minor comments to the manuscript:

- A short paragraph on the role of miRNA in human cancer in general would be of help for a better understanding of the role of miRNA in cancer generally and in TNBC in particular. Suggested references like the following may be added:

  • Peng Y, Croce C. The role of MicroRNAs in human cancer. Sig Transduct Target Ther1, 15004 (2016).
  • Stahlhut Espinosa CE, Slack FJ. The role of microRNAs in cancer. Yale J Biol Med. 2006;79(3-4):131-140.
  • Cui M, Wang H, Yao X, Zhang D, Xie Y, Cui R and Zhang X (2019) Circulating MicroRNAs in Cancer: Potential and Challenge.  Genet.10:626.
  • Si W, Shen J, Zheng H. et al.The role and mechanisms of action of microRNAs in cancer drug resistance. Clin Epigenet 11, 25 (2019). 

- Alsoa short paragraph on the differences in miRNA implications between TNBC and ER pozitive/ ER(-) breast cancer would contribute to a better understanding of the exact role of miRNA in TNBC as compared with the other types of breast cancer (see Si W et al above).

- Page 5. Paragraph 3.1. Immune cells

NETosis are a newly described mechanism in cancer. Perhaps it would be of interest to mention if there is a connections between NETosis and miRNA.

               Schultz DJ, Muluhngwi P, Alizadeh-Rad N, Green MA, Rouchka EC, Waigel SJ, et al. (2017) Genome-wide miRNA response to anacardic acid in breast cancer cells. PLoS ONE 12(9): e0184471. 

Page 12. Paragraph 5. miRNA Therapeutics Targeting....

It would be of interest to mention those clinical studies (short paragraph) where miRNA added to chemotherapeutic agents led to improvement of results in TNBC or other types of breast cancer (see Si W, et al).

Minor corrections:

References 63 should be cited according to the journal style.

Best regards

Round 2

Reviewer 1 Report

I guess that according to the modifications, the paper is now suitable for publication in IJMS